# Dopamine Modulates the Processing of Food Odour in the Ventral Striatum

**DOI:** 10.3390/biomedicines10051126

**Published:** 2022-05-12

**Authors:** Olivier Rampin, Audrey Saint Albin Deliot, Christian Ouali, Jasmine Burguet, Elisa Gry, Gaelle Champeil Potokar, Nathalie Jérôme, Olga Davidenko, Nicolas Darcel, Vincent Bombail, Philippe Andrey, Isabelle Denis

**Affiliations:** 1Université Paris-Saclay, AgroParisTech, INRAE, UMR PNCA, 75005 Paris, France; audrey.saint-albin@inrae.fr (A.S.A.D.); christian.ouali@inrae.fr (C.O.); elisa.gry9@etu.univ-lorraine.ff (E.G.); gaelle.champeil-potokar@inrae.fr (G.C.P.); nathalie.jerome@inrae.fr (N.J.); olga.davidenko@agroparistech.fr (O.D.); nicolas.darcel@agroparistech.fr (N.D.); vincent.bombail@sruc.ac.uk (V.B.); isabelle.denis@inrae.fr (I.D.); 2Institut Jean-Pierre Bourgin (IJPB), Université Paris-Saclay, INRAE, AgroParisTech, 78000 Versailles, France; jasmine.burguet@inrae.fr (J.B.); philippe.andrey@inrae.fr (P.A.)

**Keywords:** food odour, dopamine, ventral striatum

## Abstract

Food odour is a potent stimulus of food intake. Odour coding in the brain occurs in synergy or competition with other sensory information and internal signals. For eliciting feeding behaviour, food odour coding has to gain signification through enrichment with additional labelling in the brain. Since the ventral striatum, at the crossroads of olfactory and reward pathways, receives a rich dopaminergic innervation, we hypothesized that dopamine plays a role in food odour information processing in the ventral striatum. Using single neurones recordings in anesthetised rats, we show that some ventral striatum neurones respond to food odour. This neuronal network displays a variety of responses (excitation, inhibition, rhythmic activity in phase with respiration). The localization of recorded neurones in a 3-dimensional brain model suggests the spatial segregation of this food-odour responsive population. Using local field potentials recordings, we found that the neural population response to food odour was characterized by an increase of power in the beta-band frequency. This response was modulated by dopamine, as evidenced by its depression following administration of the dopaminergic D1 and D2 antagonists SCH23390 and raclopride. Our results suggest that dopamine improves food odour processing in the ventral striatum.

## 1. Introduction

The sensory properties of food, including food odour, are determinants of food intake [1,2]. Food odour plays a role in locating food, memorizing food location, and learning from others what to eat [3]. In the brain, food odour information competes with other information that has either similar, positive (vision of food) or opposite, negative (smell of a predator) meanings. Food odour may or may not elicit a feeding response depending on the internal (metabolic, energetic) status [4]. Therefore, to elicit feeding behaviour, the importance of food odour must increase through additional labelling in the brain. One way for this to occur is through the activation of reward pathways. The ventral striatum, including the olfactory tubercle, is at the crossroads of the olfactory and reward pathways [5,6]. Ventral striatum neurones respond to a variety of monomolecular odours, mixtures of odours, and natural odours [7,8,9], including food odours [10,11]. The responses of ventral striatal neurones to odours are not fixed. In an experiment of associative learning, Setlow and coll. [8] showed that neurones increased their firing activity after the odours were paired with positive or negative outcomes. A sub-population of these neurones reversed their firing activity when the association of odour and outcome (positive or negative) had been reversed. This supports the idea that olfactory coding is modulated by contextual factors. The ventral/olfactory striatum network can code for reward prediction from an olfactory stimulus, at variance with other olfactory relays such as the piriform cortex [12]. Dopamine plays a role in reward [13,14]. Midbrain dopamine neurones predict reward [15]. As the ventral striatum receives a rich dopaminergic innervation from the midbrain ventral tegmental area [16,17], we hypothesized that dopamine plays a role in enhancing food odour information in the ventral striatum. Several studies suggest that the ventral tegmental-ventral striatum dopaminergic pathway affects odour coding. Ventral tegmental area stimulation affects the olfactory response of some ventral striatum neurones in the anesthetized rat [7]. Ventral tegmental area dopaminergic neurones innervating the olfactory tubercle are involved in odour preference in mice [18]. Dopamine also acts in the ventral striatum in relation to food or food intake. Ventral striatum dopamine levels increase in rodents aware of food to come [19], in the presence of, but not access to, food [20] or eating [20,21]. Activating striatal D1 dopaminergic receptors increases food intake in mice [22]. Internal signals linked to the metabolic and energetic status may contribute to enhancing food odour signalling. In the ventral tegmental area, leptin, a satiety hormone, inhibits dopaminergic neurons activity, while ghrelin, a gastric released hormone that signals hunger, increases their activity [23]. Ghrelin also improves food odour responses in humans and increases food odour pleasantness [24]. To test the hypothesis that dopamine modulates food odour responses in the ventral striatum, we first analysed the response of single neurones in the ventral striatum to food odour, using extracellular recordings. We combined functional data with the reconstruction of an averaged 3D brain model in which we imported the localization of recorded neurones. Our results show that a population of ventral striatal neurones respond to food odour. They suggest that there exists a spatial segregation of this food–odour responsive population. In a second series of experiments, we analysed the local field potentials elicited by food odour upon administration of a combination of agonists and antagonists of D1 and D2 dopaminergic receptors. The population response of ventral striatal neurones was characterized by increased power in the beta-band frequency. This response was modulated by dopamine, as evidenced by a depression of the response following administration of D1 and D2 dopaminergic receptor antagonists. Dopamine would be a means through which food odour messaging is increased in the brain. 

## 2. Materials and Methods

### 2.1. Animals

Adult male rats (207–510 g, Janvier Labs, Le Genest-St-Isle, France) were kept in our animal facility (room temperature 21 ± 2 °C, reverse 12 h–12 h light/dark cycle, lights on at 7 p.m.). Food (rodents M25 diet, Special Diets Services, Witham, UK) and (tap) water were available ad libitum. Rats were fasted overnight before the electrophysiological recordings. Experiments were performed according to European (European directive, 2010/63/UE) and French (articles R-214-87 to R-215-10 of the Code Rural, 2013) regulations and approved by our Institution Animal Care Committee and Ministère de l’Enseignement Supérieur et de la Recherche (APAFIS#19571).

### 2.2. Surgical Procedures

Rats were fasted overnight, anaesthetized with urethane (1.2 g/kg I.P.) and supplied with atropine hydrochloride (25 mg/kg I.M.) to limit secretions. Local anesthesia of scalp and ear canals was performed using 1% xylocaine. Rats were mounted in a stereotaxic frame. Rectal temperature was monitored, and body temperature kept constant (37 °C) using a heating pad. Respiration was monitored using a thermistor placed in front of the nostrils. A catheter was placed in the jugular vein for NaCl (0.9%) administration and drug injection. The scalp was incised, and a craniotomy was performed over the olfactory tubercle and the lateral olfactory tract. 

### 2.3. Electrodes

Electrical stimulation was performed with bipolar concentric electrodes (O.D. 400 µm, resistance 10 kΩ). Electrodes were placed in the lateral olfactory tract (6.5 mm rostrally and 2 mm laterally relative to bregma, 4 mm under the cortical surface). They were connected to an insulated stimulation device (model DS2, Digitimer Ltd., Welwyn Garden City, UK) delivering square wave pulses. Single-unit recordings were performed with glass pipettes (model GC120F-10, 1.2 mm O.D., Clark Electromedical Instruments, Harvard Apparatus, Edenbridge, UK) and pulled using a vertical puller (model PE2, Narishige, Tokyo, Japan). Pipettes were filled with sodium acetate 0.6 M with 1% pontamine sky blue (BDH Laboratory supplies, Poole, UK). They had a resistance of 2 to 15 MΩ. They were connected to an electrometer. A reference electrode (silver chloride thread) was placed under the skin of the animal and connected to the ground of the electrometer. The recording electrode was then driven down vertically to the olfactory tubercle using a micro positioner (model 2662, David Kopf Instruments, Tujunga, CA, USA). Local field potentials (LFP) were recorded using tungsten concentric electrodes (O.D. 400 µm, resistance 10 kΩ, WPI, Sarasota, FL, USA). The position of the LFP recording electrode was at anterior 3 mm and lateral 2 mm relative to bregma, at a depth 0.5 mm dorsal to the site where we recorded a reversion of the potentials evoked by lateral olfactory tract stimulation. Electrophysiological signals were amplified (×1000 for single-unit recordings, ×5000 for LFP) and filtered (1 Hz–10 kHz). Respiration, recording electrode potentials, and electrical and odour stimulation signals were acquired using Spike 2 software through a micro 1401 MkII analog-to-digital interface (Cambridge Electronic Design Ltd., Cambridge, UK). 

### 2.4. Stimulations

The lateral olfactory tract was stimulated with trains of 4 square wave pulses, 1 ms duration at 10 Hz, and 4 s between trains. Odour stimulation was performed with either 3 or 10 g of rodent M25 diet pellets freshly crushed and placed in separate channels of an air-dilution olfactometer at 0.5 L/min flow using deodorized air. A control channel conveyed air only. In LFP recording sessions, rats were first exposed to 2 series of odour stimulation (3 g, 10 g, and air in a randomized order, 2 s each) at 5 min intervals, then received an intravenous injection of either NaCl 0.9% (*n* = 9 rats) or a combination of dopaminergic receptors agonists and/or antagonists (*n* = 8 rats per group). Rats were then further exposed to odour stimulation (3 g and 10 g) 5 min after injection. 

### 2.5. Drugs

All drugs were purchased from Tocris (Bio-Techne, Châtillon sur Seiche, France). The dopamine D1 receptor agonist A 68930 hydrochloride was solubilized in NaCl 0.9% and injected with 0.3 mg/kg. The dopamine D1 receptor antagonist SCH 23390 hydrochloride was solubilized in NaCl 0.9% and administered at the dose of 1 mg/kg. The dopamine D2 receptor agonist quinpirole hydrochloride was solubilized in NaCl 0.9% and injected at the dose of 0.3 mg/kg. The dopamine D2 receptor antagonist raclopride was solubilized in ethanol and administered at the dose of 0.2 mg/kg (final dilution in NaCl 0.9%). 

### 2.6. Histology

Glass pipette recording sites were labelled with ejection of pontamine sky blue with a negative current (−10 µA, 7′ on-7′ off for 35 min, Midgard precision current source, Stoelting, Wood Dale, IL, USA). Rats were euthanised with urethane (2.5 g/kg) and transcardially perfused with 200 mL of Ringer solution and 500 mL 4% paraformaldehyde. The brain was removed, postfixed (48 h in 4% paraformaldehyde), cryoprotected (72 h in 30% sucrose), and cut on a cryostat (model 3050S, Leica Microsystems, Rueil Malmaison, France) in 40 µm thick serial sections in the coronal plane. Sections were collected on slides and stained with cresyl violet acetate. To digitize sections, slides were scanned on a digital slide scanner (Nanozoomer, Hamamatsu, Massy, France). Sections were allocated to one out of six reference coronal planes from the Paxinos and Watson rat brain atlas [25]. 

### 2.7. 3D Reconstructions and Spatial Normalization

Using Free-D software [26], contours of the brain, corpus callosum, ventricles, anterior commissure, lateral olfactory tract, and lamina II of the piriform cortex and the olfactory tubercule were manually drawn on each digitized section. In addition, the position of the recorded neurone (pontamine sky blue ejection site) was marked. A 3D surface reconstruction was obtained for each section by replicating contours at a distance of ±100 µm before and after their position along the antero-posterior axis of the brain. For each of the six coronal planes, the surfaces of all structures, except the olfactory tubercle, which was too variable in shape to be considered an anatomical landmark, were registered, and averaged among individuals using the group-wise surface registration and averaging algorithm [27] available in Free-D’s spatial normalization module [28]. The deformation field between each 3D reconstruction and the average reconstruction was computed, and 3D multivariate polynomials of degrees varying between 2 and 7 were fit to this deformation field. Minimizing the residual root-mean square distance between the landmark structures after polynomial warping was used as a criterion to select the optimal degree of polynomial deformations. The corresponding polynomials were used to warp the recorded neurone position from the reconstruction of the individual slice to the average one.

### 2.8. Data Analysis

Analysis of electrophysiological signals was performed using Spike 2 software. From single-unit recordings experiments, results only include units in which (i) activity was changed by electrical stimulation of the lateral olfactory tract, and (ii) the recording site was identified by a dye (blue) labelling on brain sections. We used a 30 s period of spontaneous activity to confirm that only one neurone was recorded at a time (auto correlogram, no spike occurring within a 1.5 ms interval). This was done to calculate spontaneous activity (expressed as spikes/s) and search for modulation of spontaneous activity along the respiratory cycle (averaging the respiratory cycle over this period, then establishing a stimulus time histogram of action potentials in phase with the average respiratory cycle). The response to odour was evaluated by numbering action potentials during 2 s (in 250 ms bins) before and 2 s during odour stimulation. If the comparison between the 2 showed a statistical significance (paired Student’s *t*-test, *p* < 0.05), then the neurone was considered either activated or inhibited. If no difference was observed, we further analysed whether neurone activity became rhythmic during odour stimulation. The neurone was then considered rhythmic or with no odour effect. 

A Monte–Carlo method was designed and implemented in the R software [29] to evaluate whether, for each reference coronal plane, the responding neurones obeyed a specific, non-random spatial organization. Each neurone was assigned the label “responding” (either activated or inhibited) or “not responding”. The number N of not responding neurones contained within the convex hull of the responding neurones was determined. This number was also computed after having randomly shuffled the responding/not responding labels among the observed neurone positions. There were 39 such randomizations performed, yielding N1 to N39 numbers of not responding neurones within the convex hull of the responding neurones under a random organization. The 40 values N, N1, …, N39 were sorted in ascending order, and the rank r of the observed count N was determined. A low value of this rank indicated that responding neurones were segregated from the non-responding ones.

For LFP signals, we focused on beta (15–40 Hz) activity. Electrophysiological recordings were filtered (bandpass 15–40 Hz). The power spectra of the filtered electrophysiological signal were computed for 2 s during odour stimulation. We normalized this power by the power in a 2 s window before odour stimulation. Comparisons were performed in the different conditions using this ratio [power spectra for 2 s during odour/power spectra for 2 s before odour] as the variable.

Effects of food odour (3 g vs. 10 g vs. air), repetition (between the two series of odours before drugs injections), and drugs were searched for using non-parametric statistical analysis with repeated measures, namely Friedman’s test for three samples with Dunn’s multiple comparison test, and Wilcoxon’s test for two samples (5 min before and 5 min after drugs injection). Differences were considered statistically significant for *p* < 0.05. Statistical analysis was performed using GraphPad Prism 5 software (GraphPad Software, San Diego, CA, USA). 

## 3. Results

### 3.1. Single-Unit Recordings

Ninety-four neurones responded to the electrical stimulation of the lateral olfactory tract by a change in their spontaneous activity. Figure 1A illustrates the variety of spontaneous activity in this neuronal population. There was no correlation between spontaneous activity and the depth at which neurones were recorded (Spearman r = 0.2, *p* = 0.0559, Figure 1B). The spontaneous activity of 39 out of 94 neurones displayed some modulation in phase with the respiratory cycle (Figure 2). About half of the neurones (41/94) responded to food odour. Within this group, neurones that were activated were more numerous (*n* = 23, Figure 3A), followed by neurones that decreased their activity upon stimulation (*n* = 10, Figure 3B). These changes represented a 70% increased activity for excited neurones, and a 42% decreased activity for inhibited neurones. There was a correlation between food odour-evoked excitation and spontaneous activity (Spearman r = −0.5744, *p* = 0.0041) (Figure 4). The activity of a small neuronal population became rhythmic upon food odour stimulation (*n* = 8, Figure 3C). This change did not last for the duration of the stimulation. Finally, 53/94 neurones did not display any change (Figure 3D).

### 3.2. 3D Reconstruction

As revealed by the labelling of their recording sites, recorded neurones were distributed in the ventral striatum, including the olfactory tubercle and nucleus accumbens (Figure 5A).

To analyse the spatial distribution of neurones responding to food odour, we generated average and spatially normalized representations for the six reference coronal planes along the antero-posterior axis of the brain, using the contours of the brain, corpus callosum, ventricles, anterior commissure, lateral olfactory tract, and pyriform cortex layer II as anatomical landmarks (Figure 5B). Spatial normalization was performed using polynomial warping of degree 5, as this degree minimized the residual error between landmarks (Appendix A). The distribution of neurones following spatial normalization at each of the six coronal planes suggested that responding neurones were spatially segregated from the non-responding neurones (Figure 5C). To test this spatial segregation, we compared the observed number of non-responding neurones contained within the convex hull of the responding neurones to the expected number under a random organization of responding and non-responding neurones. Except for the posterior-most localization, where the procedure could not be applied because only one neurone was observed, we found that the observed number was significantly below that expected under randomness, confirming the existence of a spatial organization in the response to food odour (Figure 5D). The observed patterns suggested a progressive shift of the responding neurones from a lateral localization in the anterior-most part of the analysed ventral striatum region to a more central localization in its posterior-most part (Figure 5C).

### 3.3. LFP

Power spectrum analysis of LFP recordings in the ventral striatum showed that food odour stimulation elicited a significant increase in beta frequency activity for both 3 g and 10 g (F (2.21) = 12, *p* = 0.0011) crushed food pellets as compared to empty container, but there was no difference between 3 g and 10 g (1st test, Figure 6). 

A second stimulation 5 min later still elicited a significant increase in beta frequency activity for both 3 g and 10 g (F (2.21) = 10.75, *p* = 0.0024) with no difference between 3 g and 10 g (2nd test, Figure 6). On some occasions, we could simultaneously record multiunit activity and local field potentials (Figure 7). These electrophysiological recordings confirmed that food odour elicited increased rhythmic neuronal activity along with increased power in the beta frequency band. 

Intravenous administration of NaCl 0.9% had no effect on the increased activity in the beta frequency domain in response to odour released by 3 g (*p* = 1) and by 10 g crushed food pellets (*p* = 0.4187) (Figure 8A). Intravenous administration of the dopaminergic antagonists D1 (SCH23390) + D2 (raclopride) elicited a significant decrease in the beta frequency activity in response to food odour for both 3 g (a 36% decrease, *p* = 0.0078) and 10 g (a 66% decrease, *p* = 0.0156) (Figure 8B). Intravenous administration of the dopaminergic agonists D1 (A68930) + D2 (quinpirole) had no effect on the LFP increase in response to 3 g (*p* = 0.3125) and to 10 g (*p* = 0.3828) food odour (Figure 8C). 

Likewise, neither a combination of the D1 agonist A68930 + D2 antagonist raclopride (Figure 9A) nor the combination of D1 antagonist SCH23390 + D2 agonist quinpirole (Figure 9B) had any effect on the LFP increase elicited by 3 g (respectively *p* = 0.5469 and *p* = 0.0781) and by 10 g (respectively *p* = 0.3828 and *p* = 0.6406) food odour. 

## 4. Discussion

In this study, we demonstrate that some of the ventral striatum neurones present in nucleus accumbens and olfactory tubercles respond to food odour. These neurones displayed spontaneous activity spanning over a large range, in keeping with previous reports [7,8,10,11,12]. Oettl and coll. [30] considered units with a baseline firing rate < 5 Hz as striatal projection neurones. Some but not all the neurones that we recorded had spontaneous activity within this range. This suggests that we recorded a heterogeneous population, including striatal projection neurones and interneurones.

Ventral striatum neurones respond to a variety of sensory inputs: odours, sounds, touch [7,9]. The different sensory inputs interact so that, e.g., neurones activated by an odour display an increased activation when the same odour is coupled with a tone stimulus [9]. It suggests that the food odour-sensitive network we recorded may respond to other sensory inputs. Ventral striatum neurones display a variety of responses to odours. The same neurone may be activated by a single odorant molecule and inhibited by another one [7], as well as activated by complex natural odours while inhibited by others (predator, congener, food [10,11]). It suggests that population activity, as recorded through local field potentials, may bring a more informative, integrative view of the ventral striatum network response to food odour. 

In our hands, and as reported by others, there are more ventral striatum neurones excited by odours than neurones inhibited by odours [12,30,31]. When one considers neurones excited by odours, Midroit and coll. [32] showed that attractive odorants activated more neurones in the olfactory tubercle than unattractive odorants, as evidenced by Fos labelling. Additionally, only attractive odorants elicited a statistically significant rise in activity. Gadziola and coll. [33] showed that the increased activity in response to odours was further enhanced when coupled with a reward stimulus. Food odour likely represents a reward stimulus for the rats we recorded because they were overnight fasted and stimulated with food odour that was familiar to them. Therefore, the increased activity that we recorded may correspond to a “reward coupled” response. 

We recorded activated neurones during odour stimulation; therefore, we expected an increase in power in LFP recordings. We also recorded neurones that were inhibited, and this could improve the LFP signal through an increase in the signal/noise ratio. Finally, we recorded a rhythmic pattern of neuronal activity during odour stimulation, so we also expected a translation of this rhythm in LFP recordings. Interestingly, Shusterman and coll [34] suggested that a rhythmic pattern, locked to the sniff phase, in response to odour facilitates synchrony among neuronal populations independent of variations in respiration frequency.

Ventral striatum neurones respond to odours, be they monomolecular odorants, complex mixtures, or a priori new or familiar odours. The response relies on a direct excitatory innervation from olfactory bulb neurones, which axons travel in the lateral olfactory tract. Added to this strong input is modulation. Neurones that respond to a neutral odour increase their firing if the odour is paired with a reward (a sucrose solution, [8,31]). Some neurones reverse their firing selectivity if the link between an odour and the outcome is reversed (now coupling the same odour with a quinine solution, [8]). The rapid change in neuronal response to the same odour, depending on the context, likely relies upon the recruitment of different neuronal pathways that already exist and impinge onto ventral striatum neurones. Because changes in neuronal responses are modulations of activity rather than complete activation or inhibition, it suggests that local neuromodulators, such as dopamine, also play a role [35]. 

Our representation of the distribution of recorded neurones, labelled with their response in a three-dimensional space, is a first attempt to associate both functional and anatomical data at the neuronal scale in an integrated view. This gives more information than the two-dimensional representation of the population of recorded neurones (activity along time on the x-axis, individual neurones on the y-axis [12,30,31,36], or three-dimensional brain maps, e.g., those displaying Fos labelled, odour activated neurones [37]. In such maps, established post-mortem, activated only (Fos positive) neurones are localized. Functional magnetic resonance imaging (fMRI) provides real-time neuronal activations and inhibitions in the whole brain, e.g., activation maps of the rat brain during olfactory stimulation [38] but still does not provide resolution at the neuronal scale.

The neurones that we recorded and that were responsive to food odour (which likely represents a rewarding odour) were grouped in the lateral domain of the ventral striatum/olfactory tubercle. This finding contrasts with the current hypothesis on spatial organization within the ventral striatum, that the medial dopaminergic projection system, from the posteromedial ventral tegmental area to the ventromedial striatum, would be more important in reward mechanisms than the lateral projection system [16]. In rats, self-administration of amphetamine is more efficient in the ventromedial striatum (medial part of the nucleus accumbens shell and olfactory tubercle) than in the ventrolateral striatum [39]. Moreover, using Fos labelling as a proxy for neuronal activation, Murata and coll. [37] showed that the anteromedial olfactory tubercle was activated by odour cues associated with a reward (sucrose solution), while the lateral olfactory tubercle was activated by odour cues associated with a negative outcome (electric shock). The olfactory tubercle is made of three layers [40]. Most of the neurones that we recorded were present in layer three (multiform layer), suggesting that they were interneurones. These interneurones may contribute to an increase in the signal (food odour)/noise (ambient odour) ratio by inhibiting local olfactory networks not involved in food odour processing. Other recorded neurones may be output neurones whose axons innervate the pallidum. The ventral pallidum receives mesolimbic dopaminergic projections and interacts with the lateral hypothalamus, an important centre in the control of feeding [41]. The neurones that we recorded may represent the olfactory input contribution to the activation of this food intake- controlling network. 

The LFP recordings evidence increased oscillations in olfactory structures in response to odour stimulation [42]. Oscillations reflect the population’s response to the stimulus [43,44]. There are three sets of oscillations recorded in the olfactory system. Beta (15–40 Hz) and gamma (30–80 Hz) waves are induced by odour inputs [42], while theta waves (3–12 Hz) are coupled with respiration. Odour stimulation decreases gamma wave power and increases beta activity [42]. Furthermore, in freely moving rats, dry food pellet odour increases olfactory (piriform) cortex activity in the beta-band frequency [45]. These observations led us to focus on beta waves and their variations during odour presentation and following drug injection in the anesthetized rat. We recorded a significant increase of power in the beta-band frequencies upon food odour stimulation. 

We found no difference in the responses elicited by 3 g and 10 g freshly crushed food pellets, suggesting that the increased power of the beta-band frequency does not integrate odour intensity. Therefore, ventral striatum coding of food odour appears qualitative rather than quantitative. To compare the effects of food odour stimulation before and after drug injections, we checked for the effects of a second stimulation. We observed that the second set of stimulations performed 5 min after the first one still elicited power increases in the beta-band frequencies. Although we cannot exclude habituation, namely a lower response to a second stimulation, we were in a position to test the effects of drugs. 

Dopamine is a major actor in the reward circuitry [16,46], playing a part in a variety of rewards including food [47]. The presence of dopamine in the ventral striatum, including the olfactory tubercle, has long been established [48,49]. D1 and D2 dopaminergic receptors have been evidenced there [17,50,51].

There is an increase in dopamine efflux in the nucleus accumbens of fasted rats exposed to food before they gain access to it [20]. The view and odour of food are likely responsible for this early dopamine release. Food pellet consumption activates dopaminergic terminals in the olfactory tubercle, and activation of the ventral tegmental area to the olfactory tubercle dopaminergic pathway elicits the preference for a neutral odour [18]. These data suggest a direct link between dopamine and the processing of food odour by the ventral striatum. 

Fasted rats were used in this study. This condition probably increased dopamine release in the ventral striatum because ventral tegmental area dopaminergic neurones are directly activated by hunger signals such as ghrelin [52] as well as by the lateral hypothalamic feeding centre [53].

We investigated the influence of dopamine on the beta oscillations within the ventral striatum by using drugs that interact with D1 or D2 dopaminergic receptors and recorded LFP signals. Dopamine has been shown to alter LFP oscillations in other non-olfactory neural networks. The power of beta oscillations in the subthalamic nucleus is decreased by levodopa, a precursor of dopamine [54]. Recordings of dopamine release and beta-band activity in the striatum of non-human primates performing a task evidenced that the task elicited beta-band activity decreases preceding a dopamine increase [55].

Following the injection of the dopaminergic D1 and D2 antagonists SCH23390 + raclopride, we measured a decrease in power in the beta-band frequencies in response to food odour stimulation. This indicates that dopamine facilitates the response to food odour in the ventral striatum through D1 and D2 receptors. Inokuchi [56] observed that the D2 dopaminergic receptor antagonist haloperidol depressed the inhibitory effect of olfactory bulb stimulation on some olfactory tubercle neurones. Coadministration of the same D1 and D2 antagonists SCH23390 + raclopride abolished the formation of odour preference in mice [18]. Blocking the D1 dopaminergic receptor with SCH23390 was sufficient to suppress conditioned place preference elicited by an attractive odour [32]. Dopamine acts through D1 and D2 receptors at different sites along the olfactory pathways. For instance, the D2 dopaminergic receptor antagonist spiperone increases the proportion of olfactory bulb neurones that respond to an odour [57]. 

In our experiment, the coadministration of D1 and D2 antagonists did not completely abolish the response to food odour. This observation suggests that dopamine only exerts a modulation of the ventral striatum response and/or that D1 and D2 receptors display opposite roles (e.g., the above-cited increased response to odour in the olfactory bulb following D2 antagonist injection). Axons of mitral and tufted cells activated by odours release the potent excitatory amino acid glutamate in the ventral striatum. It is the major contribution to ventral striatum neurones excitation. As evidenced by others, dopamine only modulates [33] and/or reinforces [30] this activation.

Coadministration of the dopaminergic D1 and D2 agonists A68930 and quinpirole had no effect on the power increase of beta-band frequencies elicited by food odour stimulation. Berke [58] recorded a weak or no effect of the D1 and D2 dopamine agonist apomorphine and the indirect, non-selective dopamine agonist amphetamine on LFP oscillations < 40 Hz in the striatum of rats. It may be that in our condition, the local dopaminergic system is already activated, partly saturated by local dopamine released by the fasted status of the rats. Or it may be that D1 and D2 dopaminergic receptors have opposing effects in our model, leading to a null effect of coadministration. In accordance with this hypothesis, Doty and Risser [59] have shown that the D2 agonist quinpirole decreases odour detection performance in rats. 

To better disentangle the contribution of D1 and D2 receptors in the LFP power increase to food odour stimulation, we used coadministrations of the D1 agonist A68930 and D2 antagonist raclopride, then of the D1 antagonist SCH23390 and D2 agonist quinpirole. None of these combinations had any effect on the signals. 

If D1 and D2 receptors act in synergy, then using antagonists to both should affect LFP power increase to food odour. This was observed following the coadministration of the two antagonists SCH23390 + raclopride. However, in an experiment of odour discrimination learning, Yue and coll. [60] showed that D1 and D2 dopaminergic receptors had opposing effects; a D1 agonist and a D2 antagonist facilitated the discrimination of similar odours–furthermore, dopamine targets receptors at multiple sites of the olfactory brain. When injected into the olfactory bulb, only the D2 receptor agonist quinpirole and the D2 antagonist sulpiride affected discrimination performance [61]. By delivering D1 and D2 agonists and antagonists intravenously and recording ventral striatum activity, we were not able to solve this complexity, likely due to different effects at multiple sites. 

## 5. Conclusions

The olfactory tubercle encodes odour valence [31]. The role of the olfactory tubercle in odour-guided eating behaviour has recently been hypothesized [62]. Our contribution to this hypothesis is that, within the lateral ventral striatum, there exists a neuronal population responding to food odour by an increased (main response) or decreased activity. Food odour also increases the power of the beta oscillations, and this response is modulated by dopaminergic innervation from the tegmental area. This is the first electrophysiological evidence that dopamine contributes to the coding of neuronal responses to food odour in the ventral striatum. Our approach is complementary to those performed in humans [63], targeting a better understanding of the brain processing of food-related odours leading to appetitive behaviour, both in normal and obese patients [64]. In obese patients, an augmented sensitivity to food-related odours, likely through changes in the dopaminergic reward pathway, is recorded [64]. With regards to neurodegeneration of dopaminergic pathways, such as that reported in, e.g., parkinsonism, deleterious effects are likely due to the decrease of a direct effect of dopamine on odour coding pathways, to which decreased dopaminergic activation of reward/hedonic pathways is added. 

## Figures and Tables

**Figure 1 biomedicines-10-01126-f001:**
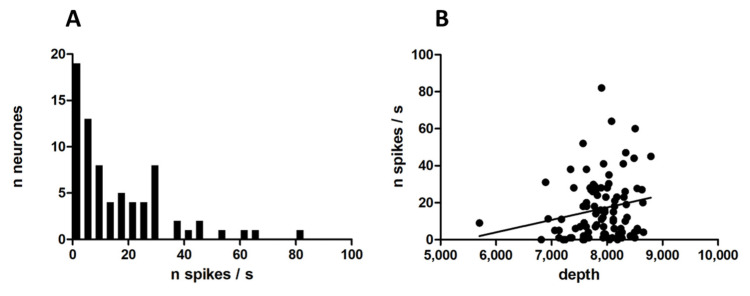
Effects of food odour on single neuronal recordings in the ventral striatum. (**A**). Number of neurones as a function of spontaneous activity (bin width: 4 spikes/s). Median, 11 spikes/s, range, 0–82 spikes/s. (**B**). Relation between spontaneous activity (spikes/s) and depth of recording (micrometers). Each point displays the mean spontaneous activity of a neurone plotted versus the depth under the cortical surface at which it was recorded. There was no correlation between these 2 variables.

**Figure 2 biomedicines-10-01126-f002:**
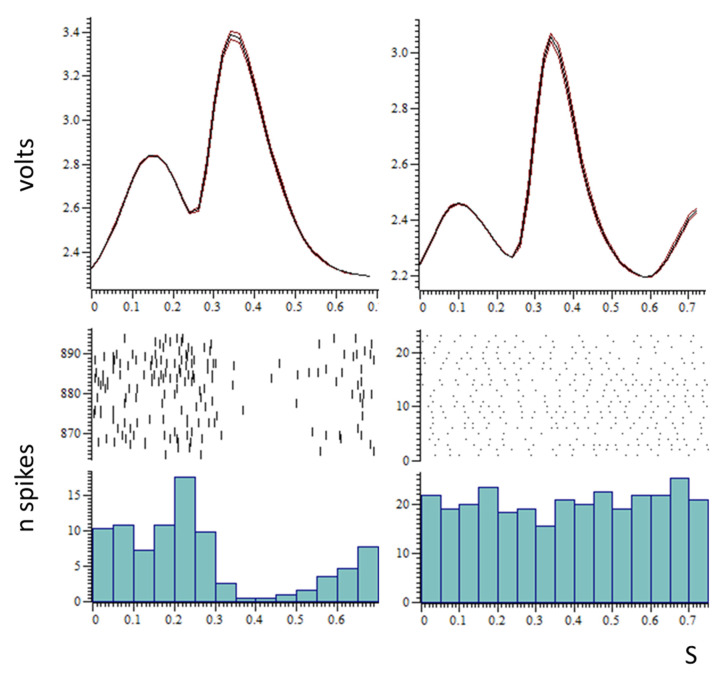
Examples of spontaneous activity in ventral striatal neurones in phase with (**left column**), or independent of (**right column**), the respiratory cycle. Top graphs: average (mean +/− SEM) of a respiratory cycle computed over 30 s, inspiration (1st part of the curve, low amplitude) then expiration (2nd part of the curve, high amplitude). Bottom graphs: upper part, single neurone raster and lower part, peristimulus time histograms (bin width 50 ms) of spontaneous activity (n spikes) computed over the same 30 s period. X axis time in seconds. Left column: this neurone became nearly silent during expiration. Right column: this neurone’s spontaneous activity was independent from respiration.

**Figure 3 biomedicines-10-01126-f003:**
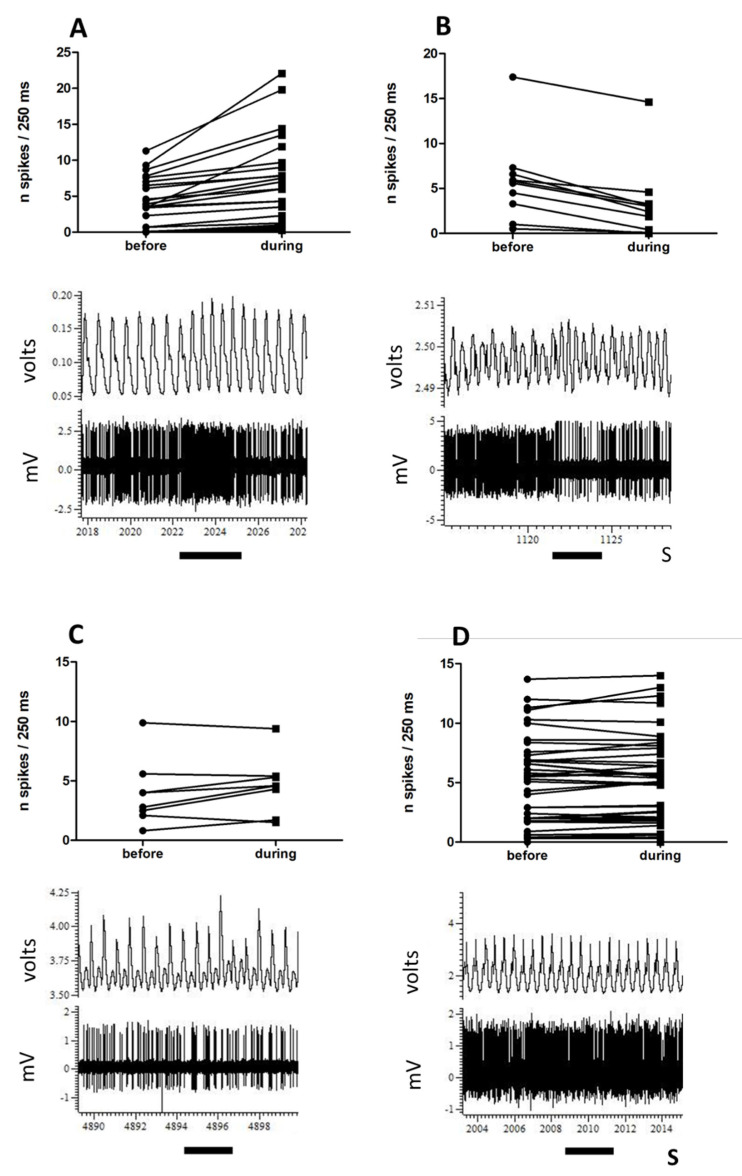
Four groups of neurones are distinguished from their response to food odour. From (**A**) to (**D**) top: all neurone activity before (**left**) and during (**right**) food odour stimulation. Activity is measured as number of spikes per 250 ms and averaged over 2 s before and 2 s during odour stimulation. Bottom: illustration of the response of a single neurone from the same group. Black bar, odour stimulation. Top trace is respiration, bottom trace is single unit recording. X axis in seconds (S). (**A**): 23 neurones increased their activity during food odour stimulation. (**B**): 10 neurones decreased their activity upon food odour stimulation. (**C**): these 8 neurones did not display a change in their activity but became rhythmic in phase with the respiratory cycle during odour stimulation. (**D**): 53 neurones didn’t respond to food odour stimulation.

**Figure 4 biomedicines-10-01126-f004:**
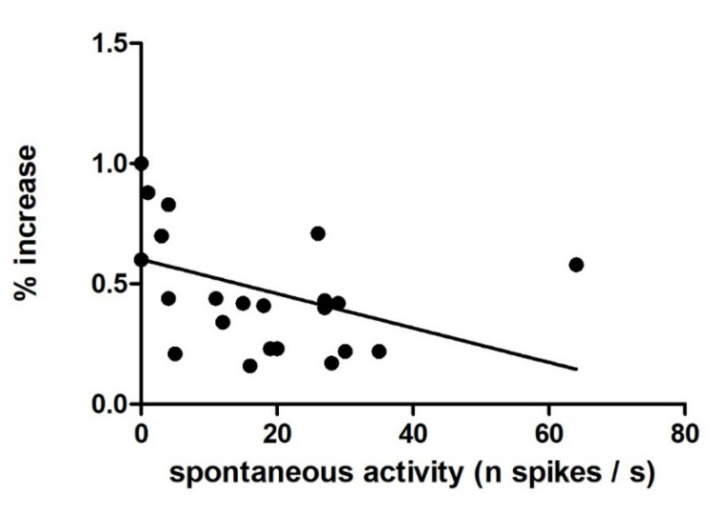
Relation between spontaneous activity of neurones and food odour evoked activation. For those 23 food odour excited neurones of Figure 3A, each point displays the percent increased activity upon food odour stimulation (y-axis) plotted versus the mean spontaneous activity (x-axis). There is a significant correlation between the 2 variables (Spearman r = −0.5744, *p* = 0.0041).

**Figure 5 biomedicines-10-01126-f005:**
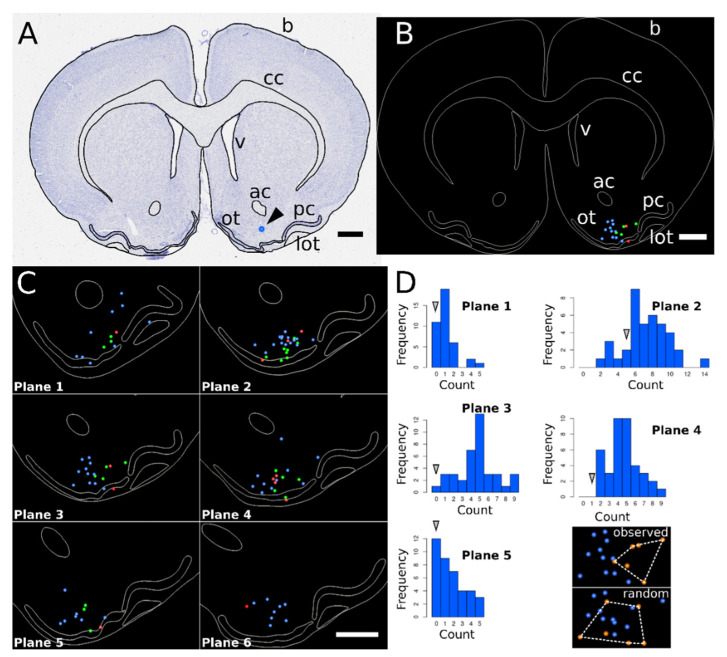
Spatial normalisation of brain coronal planes. (**A**): brain structures were delineated on all digitized brain coronal sections (contours in black; b: brain; cc: corpus callosum; v: ventricle; ac: anterior commissure; lot: lateral olfactory tract; pc: pyriform cortex, layer II; ot: olfactory tubercle, layer II) and recording site pointed (arrowhead). (**B**) All segmented structures except ot were used as reference structures for spatial normalisation of brain sections at each considered coronal plane. Using this normalisation procedure, the positions of recorded neurones were mapped into an average representation of the considered plane (illustration corresponding to Plane 3). Non-responsive, inhibited, and activated neurones are represented by blue, red, and green dots, respectively. (**C**) Enlarged portions of the ventral quadrant of six identified coronal planes, from the anterior most 1 to the posterior most 6. (**D**): quantitative analysis of the distribution of responsive neurones (red and green dots in (**C**)) within the population of all recorded neurones. The number of non-responsive neurones falling inside the convex hull of responsive neurones (corresponding bin indicated by arrows, e.g., bin 0 for Plane 1) was compared to the distribution of this quantity under random assignment of responsive/non-responsive labels to neurone positions (histograms, 39 random shuffles). Scale bars: 1 mm.

**Figure 6 biomedicines-10-01126-f006:**
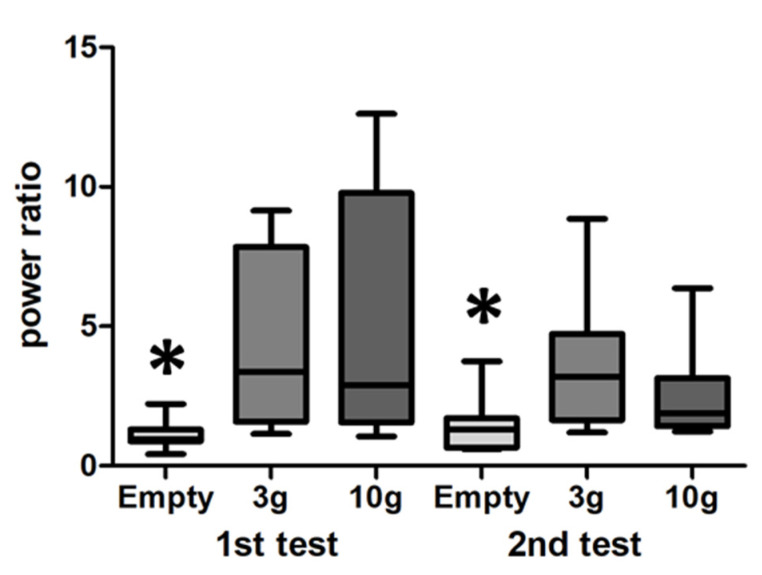
Effects of food odour stimulation on LFP recordings. (**Left**) In a first test, we observed that odour released from 3 g and 10 g of freshly crushed food pellets increased power in the beta-band frequency (15–40 Hz, power ratio: [power during the 2 s of stimulation/power for 2 s before the stimulation]) as compared to no odour (empty container) (*n* = 9 rats). A second test of odour stimulation (**right**) performed 5 min later, still elicited a power increase with 3 g and 10 g food odour stimulation, although of lesser amplitude. Empty container different from 3 g and 10 g, * *p* < 0.05.

**Figure 7 biomedicines-10-01126-f007:**
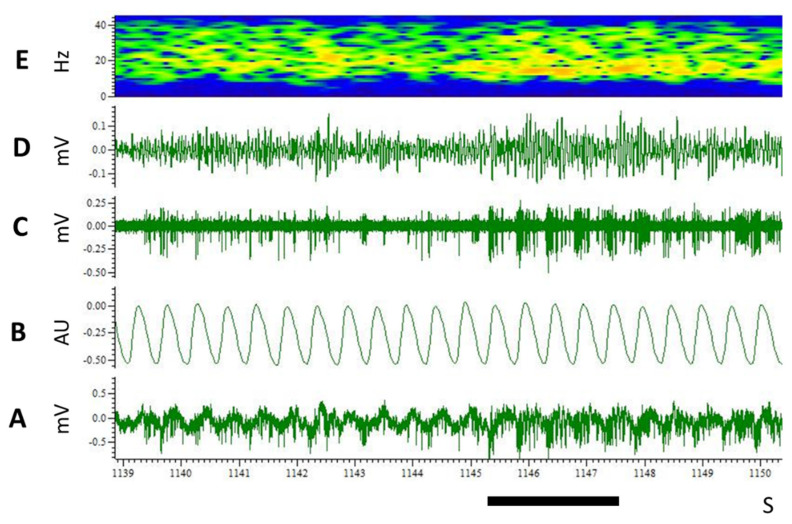
Simultaneous recordings of multiunit activity and local field potentials in the ventral striatum. The raw electrophysiological signal ((**A**), 1 Hz–10 kHz) was filtered to provide multiunit activity ((**C**), high pass > 500 Hz) and LFP in the beta-band frequency ((**D**), bandpass: 15–40 Hz). (**E**) represents the time-frequency plot of (**D**) channel (yellow-orange colours: higher power). Black horizontal bar: 2 s of food odour stimulation. (**B**): respiration. Food odour stimulation elicited increased neuronal activity, which became rhythmic [(**C**)], greater amplitude and frequency of the LFP signal (**A**,**D**), and increased power in the beta-band frequency [(**E**)]. mV millivolt; AU arbitrary units; Hz: hertz, s: seconds.

**Figure 8 biomedicines-10-01126-f008:**
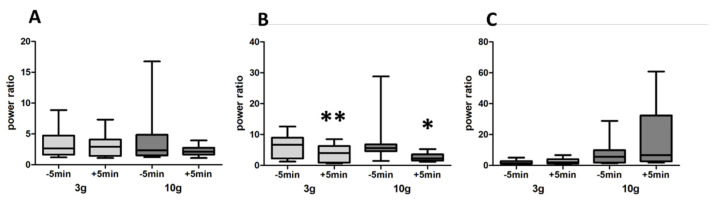
(**A**). Intravenously injected NaCl had no effect on the beta-band power elicited by 3 g or 10 g food pellet odour. Olfactory stimulations were performed 5 min before and 5 min after injection (*n* = 9 rats). (**B**). Simultaneous administration of the dopaminergic antagonists D1 (SCH23390) + D2 (raclopride) (i.v.) elicited a significant decrease of beta frequency activity to food odour for both 3 g and 10 g (*n* = 8 rats). Olfactory stimulations were performed 5 min before and 5 min after injection. (**C**). Intravenous administration of the dopaminergic agonists D1 (A68930) + D2 (quinpirole) (i.v.) had no effect on LFP increase to 3 g and to 10 g food odour (*n* = 8 rats). Olfactory stimulations were performed 5 min before and 5 min after injection. * *p* < 0.05; ** *p* < 0.01.

**Figure 9 biomedicines-10-01126-f009:**
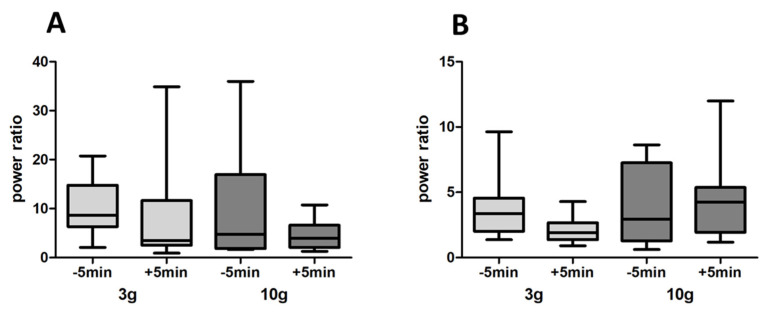
(**A**)**.** Simultaneous i.v. administration of the D1 agonist A68930 and the D2 antagonist raclopride had no effect on the LFP power increase elicited by 3 g and 10 g food odour. Olfactory stimulations were performed 5 min before and 5 min after injection (*n* = 8 rats). (**B**). Simultaneous i.v. administration of the D1 antagonist SCH23390 and the D2 agonist quinpirole had no effect on LFP power increase elicited by 3 g and 10 g food odour. Olfactory stimulations were performed 5 min before and 5 min after injection (*n* = 8 rats).

## Data Availability

Histological data are to be archived in a public dataset, the address of which will be given to you as soon as possible. Electrophysiological data are archived in a (our) laboratory-based data center. Because we are on the way to move our laboratory geographically (and administratively), we are not able to provide archives of this experimental part at this time. However, these data will be made publicly available through a publicly archived dataset as soon as the rules of our future lab and administration will be signed. Please let us know if this is a concern.

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
