# Peer review of "Dopamine Modulates the Processing of Food Odour in the Ventral Striatum"

_biomedicines, 2022, doi:10.3390/biomedicines10051126_

Round 1

Reviewer 1 Report

Rampin et al found that neurons in the ventral striatum responded to food odor with a variety of responses (excitation, inhibition, rhythmic activity in phase with respiration or no response) in rats. They showed spatial segregation of this food odor responsive population. By focusing the beta-band frequency with administration of the dopaminergic D1 and D2 agonists and antagonists, they found the neural modulation to food odor by them.

With great techniques of neuron recordings and spatial mapping, authors showed the dopamine neuron contribution to food odor responses in the ventral striatum, but additional experiments are needed to say the specificity of this population to “food odor” and contribution of dopamine to the neurons to food odor in the ventral striatum directly.

Major comments

  1. The authors used only food odor for all experiments, and said the conclusion of the contribution of the dopamine neurons in the ventral striatum to the food odor responses. Also they focused on the ventral striatum and dopamine because these are related to the reward system. However, they used only food odor to say the conclusion. They should use other odorants (neutral odorants and aversive odorants) for all experiments. If the same neurons respond to not only food odor, but also aversive odorants, the conclusion should be different.
  2. How many animals did they use for figure 6,8, and 9?
  3. In figure 6, 8, and 9, they used 3g and 10g crushed food. What is the meaning of 3g and 10g food? How many grams of food do rats eat for 1 hour? And why did they use “crushed” food?
  4. In figure 6, 8, and 9, they measured the beta frequency activities. What is the meaning of beta frequency activities to convey odor information compared with gamma and theta waves? What happened in gamma and theta waves in figure7?
  5. They used intravenously injection of dopaminergic D1 and D2 agonists and antagonists. It is true that dopamine somehow modulates the responses to food odor in the ventral striatum. But it is not clear if the dopamine directly modulates the responsive neuron to food odor in the ventral striatum or not. Can they do any other experiments to say dopamine directly modulates the neuron to food odor in the ventral striatum?

Minor comments

  1. Something strange happens in sentences of 3.Results 3.1. Single-unit recordings maybe because of the layout. The words are not connected between lines 213 and 230.

Author Response

Reviewer 1:

We thank the reviewers for their comments on our manuscript.

Major comments

  1. odorants that activate ventral striatum / olfactory tubercle neurones: this is an important aspect. Olfactory tubercle is a small olfactory relay structure as compared to the piriform cortex, both in term of size and number of neurones. So, in face of the vast number of odours and odorant molecules, one cannot expect olfactory tubercle neurones to be highly selective, i.e., tuned to a limited number of related odours or odorant molecules, such as glomeruli of the olfactory bulb, nor analytic, such as the anterior piriform cortex. Confirmation is provided by publications from other authors and us showing that olfactory tubercle neurones (i) respond to different sensory modalities (touch, sounds, odours: see refs. 7 and 9 of this revised version of our manuscript), (ii) respond to several distinct natural, complex odours (food, predator, congener, see refs. 10 and 11). It is likely that among the neurones that we recorded in the present work some neurones respond to other odours. However, comparing the response of ventral striatum / olfactory tubercle neurones to different odours was beyond the goal of our study. Having already contributed to this field (ref. 11), we selected a given olfactory input, that of food, and tested the effects of dopamine. We remain careful in concluding only on the effects of dopamine on food odour and not on other odours.
  2. The number of animals has been added in figures 6, 8 and 9 and in the methods section, page 3 lines 124 and 125.
  3. Growing and adult rats display a dietary intake of 15 g / day, i.e., an average 0.63 g / hour but in distinct, separate meals that occur more often in the dark period. Following an overnight fast, rats can display a 5 g food intake within one hour. By using 3 g and 10 g of food pellets, we estimated that we presented the lower and upper amounts of food that a rat consumed in a single meal after an overnight fast. We used freshly crushed food as a mean to facilitate the release of food odours.  
  4. Rhythmic activity in olfactory pathways (ref 41) has been separated into: theta waves (3 – 12 Hz) linked to respiration, and beta (15 – 40 Hz) and gamma (30 – 80 Hz) waves, the latter two induced by odour stimulation. Because odour stimulation decreases gamma waves while at the same time increases beta waves (ref 41), we focused on beta waves. In figure 7, channel A (bottom) has a large bandpass (1 Hz – 10 kHz) and displays all waves, including theta, beta and gamma. Channel D is a copy of A but has a narrow bandpass (15 - 40 Hz) and therefore only displays beta waves.
  5. It is true that dopamine acts at different levels of the olfactory pathways. We refer to the experiments of Wilson and Sullivan (1995, ref. 58) to illustrate the effects of dopamine on mitral and tufted cells of the olfactory bulb. Both mitral ad tufted cells send axons to the olfactory tubercle, so, among the effects of dopamine agonists and antagonists that we report in the present manuscript, some may be attributed to local, olfactory bulb effect. Local administration of dopamine agonists and antagonists into the ventral striatum would resolve this ambiguity, but at the cost of electrophysiological recordings stability and quality, so we did not consider this possibility.   

Minor comments

According to the “Instructions to authors” guide, “Figures should be placed in the main text near to the first time they are cited”, and we complied with this rule. A consequence is that figures occur in the main text, and this sometimes makes reading difficult. In page 5 of our manuscript, results section, chapter 3 paragraph 3.1. Single-unit recordings, main text occupies lines 206-209, then lines 218-221, then… page 6 lines 232-234 and page 7 lines 251-255.

Reviewer 2 Report

Rampin et al submitted their manuscript entitled "Dopamine modulates the processing of food odour in the ventral striatum" which described the electrophysiological activities in ventral striatum during odour stimulation. And they further demonstrated the importance of dopamine in this process. This is an interesting and important article indicating the role of dopamine in odour sensation, which may also explain the early symptoms of loss of smelling in some neurodegenerative condition such as Parkinson's disease. 

The script is well written and structured. It would be appreciated If authors can  include some of their opinions in the discussion part. For example, what kinds of the neurons in the ventral striatum are stimulated in their study? What's the implications of their findings for patients with obese or neurodegeneration as they mentioned in conclusion? These information will trigger some of the research directions in the future studies.

Author Response

Reviewer 2:

We thank the reviewers for their comments on our manuscript.

We added a comment on the putative role of the ventral striatum / olfactory tubercle neurones that we stimulated, in the discussion section, pages 12 and 13, lines 438-446, as well as a reference (Castro and coll., Front Syst Neurosci 2015, ref. 41).

Re. obese and parkinsonian patients, we added a comment in the conclusion section, making the results of Han et al. (Obesity, 2021, ref. 66) more explicit. 

Round 2

Reviewer 1 Report

Although they did not do additional experiments that I wanted them to do, they clearly described the reasons and answered all questions.